# Dim light at night unmasks sex-specific differences in circadian and autonomic regulation of cardiovascular physiology
Abhilash Prabhat [1] ✉, Dema Sami[1], Allison Ehlman[1], Isabel Stumpf[1], Tanya Seward[1], Wen Su[1], Ming C. Gong[1], Elizabeth A. Schroder [1,2] & Brian P. Delisle [1] ✉

Shift work and artificial light at night disrupt the entrainment of endogenous circadian rhythms in physiology and behavior to the day-night cycle. We hypothesized that exposure to dim light at night (dLAN) disrupts feeding rhythms, leading to sex-specific changes in autonomic signaling and day-night heart rate and blood pressure rhythms. Compared to mice housed in 12-hour light/12-hour dark cycles, mice exposed to dLAN showed reduced amplitudes in day-night feeding, heart rate, and blood pressure rhythms. In female mice, dLAN reduced the amplitude of day-night cardiovascular rhythms by decreasing the relative sympathetic regulation at night, while in male mice, it did so by increasing the relative sympathetic regulation during the daytime. Time-restricted feeding to the dim light cycle reversed these autonomic changes in both sexes. We conclude that dLAN induces sex-specific changes in autonomic regulation of heart rate and blood pressure, and time-restricted feeding may represent a chronotherapeutic strategy to mitigate the cardiovascular impact of light at night.

Industrialization, artificial light, and technological advances have led to global changes in lifestyles and work schedules worldwide[1–4]. These changes include increased shift workers, remote workers whose schedules match time zones in different parts of the world (e.g., call center workers), and the widespread use of artificial light and light-emitting devices at night[5,6]. Increased light exposure at night disrupts the naturally occurring light-dark cycles that entrain circadian rhythms in physiology to predictable changes in the environment[7]. Even relatively low or dim light at night (dLAN) has been shown to disrupt the mammalian circadian system. Repeated exposure to increased light levels at night decreases the amplitude of the day-night rhythms in feeding behavior, sleep-wake cycles, and, at the molecular level, circadian clock gene expression in the suprachiasmatic nucleus (SCN) of the hypothalamus and peripheral tissues[5,7,8]. In other words, relatively recent technological advances have led to increased and widespread light exposure at night, and this exposure to light at night disrupts circadian entrainment to the natural light-dark cycle to impact daily rhythms in behavior and physiology.

Studies show that occupations and lifestyles that disrupt the circadian system by increased light exposure at night are associated with an increased risk for cardiovascular disease[9–11]. Cardiovascular disease remains a leading cause of death, and many cardiovascular variables (e.g., heart rate and arterial blood pressure) have day-night rhythms that cycle with a periodicity of ≈24 h[12]. The impact that light at night has on day-night rhythms in heart rate and blood pressure is only now being understood[13–15]. Data suggest that light at night decreases the amplitude of day-night heart rate and blood pressure rhythms in people and small animals but through distinct mechanisms[14]. Light at night increases nighttime sympathetic signaling in people to reduce the amplitude of daily heart rate and blood pressure rhythms. In contrast, light at night in rats decreases nighttime sympathetic signaling to decrease the amplitude of these rhythms.

Studies show that low or dim light levels at night dampen the amplitude of rhythmic feeding behavior and activity in mice[7,16]. We found that rhythmic feeding behavior modifies the amplitude and acrophase of the day-night heart rate and blood pressure rhythms by altering autonomic signaling[17,18]. This raises the possibility that light at night may disrupt autonomic regulation of heart rate and blood pressure by decreasing rhythmic feeding behavior.

We tested the hypothesis that time-restricted feeding could normalize day-night heart rate and blood pressure rhythms in mice exposed to dLAN. First, we confirmed that dLAN disrupted day-night feeding, heart rate, and blood pressure rhythms in mice housed in thermoneutrality. We studied mice in thermoneutral housing to limit confounds associated with cold-induced sympathetic nervous system activation[19–22]. We adopted a strategy to estimate day-night rhythms in autonomic regulation of heart rate by measuring intrinsic heart rate and core body temperature changes across the

[1]Department of Physiology, University of Kentucky, Lexington, USA. [2]Department of Internal Medicine, University of Kentucky, Lexington, USA.
✉ e-mail: apr288@uky.edu; brian.delisle@uky.edu

24-hour cycle[17,23,24]. We also determined how dLAN and time-restricted feeding to the dLAN cycle impacted blood pressure and activity rhythms.

We found that dLAN decreased the amplitude of the day-night heart rate rhythm in female and male mice through distinct autonomic mechanisms. In female mice, dLAN decreased relative nighttime sympathetic regulation, but in male mice, dLAN increased the relative daytime sympathetic regulation. dLAN also increased daytime blood pressure in male mice and reduced blood pressure dipping in both female and male mice. Time-restricted feeding to the dim light cycle normalized autonomic regulation of heart rate, daytime blood pressure in male mice, and blood pressure dipping.

## Results

### dLAN reduced day-night feeding and heart rate rhythms in mice housed in thermoneutrality

Previous dLAN studies in mice were conducted at room temperature, where a significant portion of metabolism is dedicated to generating heat for thermoregulation[21,22]. This increased thermogenic demand elevates food intake and autonomic signaling, potentially affecting the impact of dLAN on day-night feeding and heart rate rhythms. As a first step, we tested whether exposing mice housed at thermoneutrality to dLAN would decrease the amplitudes of day-night feeding and heart rate rhythms similar to previous animal studies at room temperature[14].

**Feeding behavior.** The effects of dLAN on feeding rhythms in mice were studied using automated feeders. Mice were acclimated to automated feeders under 12-hour light (day) and 12-hour dark (night) cycles. The dark cycle was then switched to a dim light cycle to study the effects of dLAN. Figure 1A shows the mean food pellet intake measured every 2 h from female mice under 12-hour light/12-hour dark cycles and after exposure to dLAN ($n$ = 5). Mice under 12-hour light/12-hour dark cycles consumed most of their food at night, showing a significant day vs. night difference (percent food consumed: day- 21% vs. night- 79%). Exposure to dLAN increased food intake during the daytime (percent food consumed: day- 42% vs. night- 58%) but did not increase total 24-hour food intake (Fig. 1B). A change in day-night rhythms may reflect changes in the amplitude and phase of the 24-hour rhythm. Cosinor analysis indicated that dLAN reduced the difference in day vs. night food intake primarily by decreasing the amplitude of the 24-hour rhythm (Fig. 1B).

**Heart rate and core body temperature.** The impact of dLAN on heart rate and core body temperature rhythms was determined using telemetry (Fig. 2A). Mice were kept under 12-hour light/12-hour dark cycles and then exposed to dLAN. Figure 1C shows the three-day time series of hourly mean heart rate and core body temperature data from female mice in 12-hour light/12-hour dark cycles and after exposure to dLAN for one week. Under 12-hour light/12-hour dark cycles, heart rate and core body temperature were lower during the day and higher at night (Fig. 1D,H). Switching to dLAN eliminated the difference between day and night heart rate but not core body temperature. Cosinor analysis showed that dLAN decreased the day vs. night difference in heart rate by reducing the amplitude of the 24-hour heart rate rhythm (Fig. 1E). Cosinor analysis also identified a trend towards a smaller amplitude in core body temperature rhythms (Fig. 1I). dLAN did not change the mesor (mean) of daily heart rate and core body temperature rhythms compared to mice in 12-hour light/12-hour dark cycles (Fig. 1F,J). Compared to mice in 12-hour light/12-hour dark cycles, there was a trend for dLAN exposure to cause an earlier acrophase in the daily heart rate rhythm, resulting in an earlier acrophase in the daily core body temperature rhythm (Fig. 1F,J). The impact of dLAN on heart rate and core body temperature rhythms was also determined in male mice using telemetry (Supplementary Fig. 1). We observed a qualitatively similar trend in dLAN's impact on the amplitude of the daily heart rate rhythm in male mice (Supplementary Fig. 1C). A notable difference between female and male mice was that exposure to dLAN did not advance the acrophase of the daily heart rate or

core body temperature rhythm in male mice (compare Fig. 1F,J with Supplementary Fig. 1E,I).

**Intrinsic heart rate.** We tested if dLAN altered heart rate by affecting intrinsic heart rate (i.e., heart rate after autonomic receptor inhibition) using telemetry (Fig. 2A). We measured the intrinsic heart rate in female and male mice under 12-hour light/12-hour dark cycles and after exposure to dLAN by injecting mice with atropine and propranolol at ZT 6-7. The intrinsic heart rate measured in the mice was generally higher in mice housed in 12-hour light/12-hour dark cycles and after exposure to dLAN than the pre-injection heart rate. Exposure to dLAN did not change the intrinsic heart rate compared to the intrinsic heart rate measured from the mice when they were housed in 12-hour light/12-hour dark cycles (Supplementary Fig. 2).

These data demonstrate that exposure to dLAN decreased the amplitude of the day-night rhythms in food intake and heart rate in mice housed at thermoneutrality. The reduced day-night rhythms are secondary to decreased amplitudes of the 24-hour rhythms. dLAN exposure did not alter intrinsic heart rate, indicating that dLAN likely modified heart rate rhythms by affecting autonomic signaling.

### Time-restricted feeding to the dLAN cycle increased the amplitude of the day-night rhythms in heart rate and core body temperature

Feeding behavior in mice alters autonomic signaling and core body temperature to regulate the amplitude of the 24-hour heart rate rhythm[17]. We hypothesized that time-restricted feeding aligned with the dLAN cycle would increase the amplitudes of 24-hour heart rate and core body temperature in mice after exposure to dLAN. Mice were implanted with telemetry devices to continuously record heart rate and core body temperature. We conducted a second dLAN study that had three separate phases (Fig. 2B):

1. Mice were housed in a 12-hour light/12-hour dark cycle with unrestricted food access.
2. Mice were then exposed to dLAN with unrestricted food access.
3. Lastly, mice continued in dLAN but with feeding restricted to the dLAN cycle (ZT 12 to ZT 0).

**Heart rate and core body temperature.** Figure 3A–F shows the seven-day time series of hourly mean heart rate and core body temperature data from female and male mice. Cosinor analysis revealed that the amplitude of the 24-hour heart rate and core body temperature rhythms were reduced in mice exposed to dLAN with unrestricted food compared to those in the 12-hour light/12-hour dark cycle (Fig. 3G,J). Restricting feeding to the dLAN cycle increased the amplitudes of the 24-hour heart rate and core body temperature rhythms similar to levels observed under the 12-hour light/12-hour dark cycle.

Sex-specific differences existed in the mesor (Fig. 3H,K) and acrophase (Fig. 3I,L) of the 24-hour heart rate and core body temperature rhythms. When compared to male mice:

(1) The mesor of heart rate rhythms when housed under the 12-hour light/12-hour dark cycle and when the mice were exposed to dLAN with time-restricted feeding to the dLAN cycle were higher in female mice (Fig. 3H).
(2) The mesor of core body temperature rhythms was higher in female mice when housed under the 12-hour light/12-hour dark cycle and when the mice were exposed to dLAN with unrestricted food access (Fig. 3K).
(3) When female mice were exposed to dLAN with unrestricted food access, the acrophase in heart rate (Fig. 3I) and core body temperature rhythms (Fig. 3L) occurred at an earlier time of day.

### Quantification of day-night rhythms in autonomic regulation of heart rate

The data suggest that exposure to dLAN in mice housed in thermoneutrality unmasked sex-specific differences in autonomic regulation of heart rate.

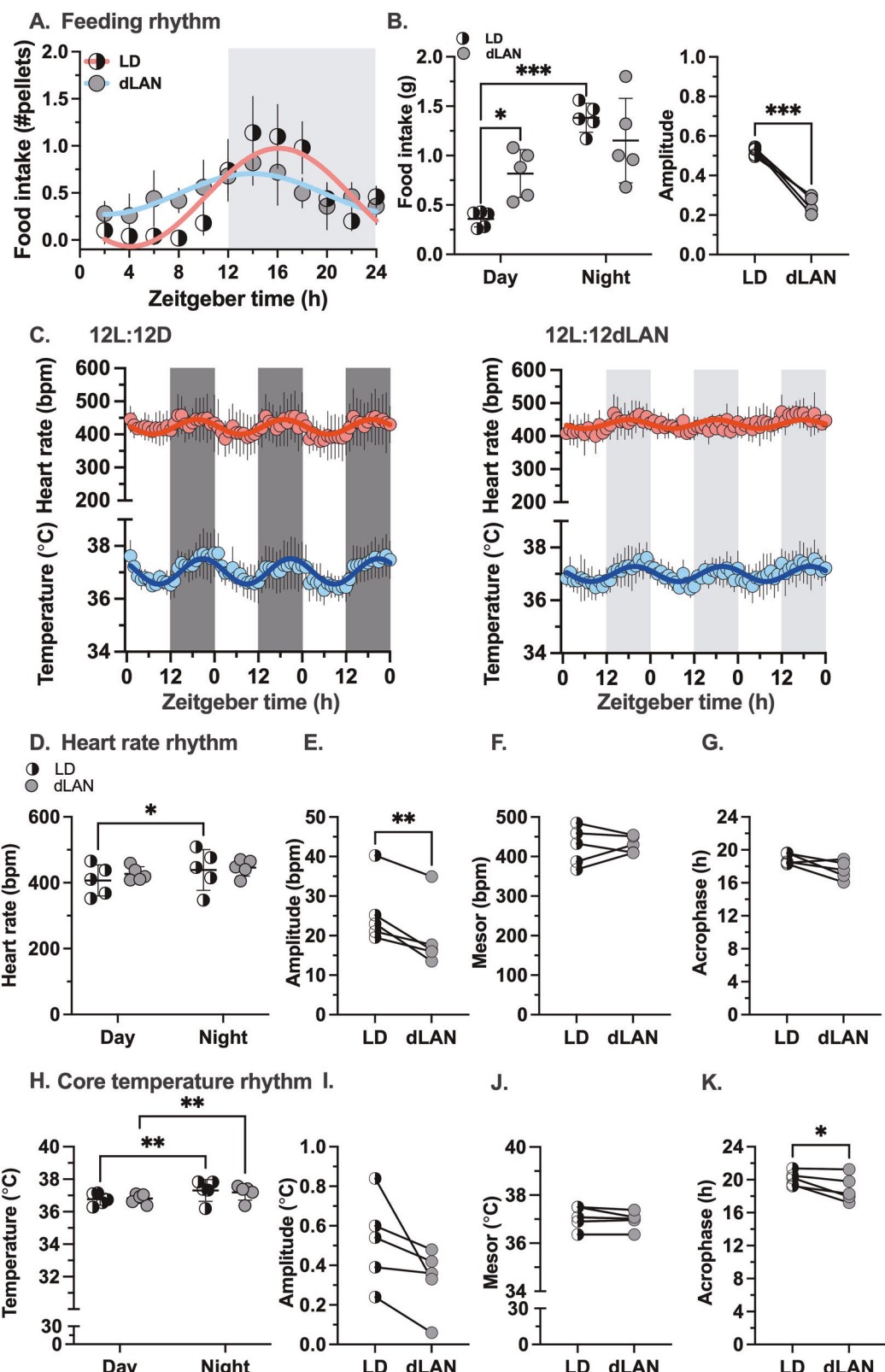

To quantify these differences, we applied the strategy of quantifying autonomic regulation of heart rate by subtracting the intrinsic heart rate from the experimentally measured heart rate.

**Intrinsic heart rate and core body temperature.** This approach requires quantifying the in vivo intrinsic heart rate and how it changes as

a function of the day-night core body temperature rhythm. To address this, we measured heart rate and core body temperature in mice housed in 12-hour light/12-hour dark cycle for two days before autonomic receptor inhibition (baseline), two days during inhibition (via serial injections with methylatropine and propranolol), and two days after inhibition (recovery) in female and male mice ($n$ = 4–5 per sex) (Figs. 2C, 4,

**Fig. 1 | dLAN reduces the amplitude of feeding and heart rate rhythms in mice housed in thermoneutrality. A** The mean food intake measured every two hours from female mice housed in LD (12-hour light: 12-hours dark; 200 lux: 0 lux, half-filled circles) or dLAN (12-hour light: 12-hour dim light at night; 200 lux: 5 lux, gray circles) were plotted as a function of zeitgeber time. The LD (red) and dLAN data (blue) were fitted with a cosine function. **B** The average food intake data was measured for each mouse housed in LD (half-filled circles) or dLAN (gray circles) during the day or night, and the amplitude of the day-night feeding rhythms was calculated using a cosine fit to the individual mouse data for each condition. **C** The hourly mean heart rate (HR, red) or core body temperature (Tb, blue) data measured from female mice housed in LD or dLAN were plotted as a function of zeitgeber time. The mean data were fitted with a cosine function (red and blue lines). **D** The average HR

was measured for each mouse housed in LD (half-filled circles) or dLAN (gray circles) during the day or night. **E–G** The amplitude, mesor, and acrophase of the day-night rhythms in HR were calculated using cosine fit to the individual mouse data for each condition. **H** The average Tb was measured for each mouse housed in LD (half-filled circles) or dLAN (gray circles) during the day or night. **I–K** The amplitude, mesor, and acrophase of the day-night rhythms in Tb were calculated using cosine fit to the individual mouse data for each condition. A paired t-test was used to compare the differences between the amplitudes. 2-way repeated measures ANOVA followed by Sidak's posthoc test for LD and dLAN day vs. night comparisons. $p < 0.05$ is considered statistically significant. $*p < 0.05$, $**p < 0.01$ and $***p < 0.001$.

Supplementary Fig. 3). We calculated the mean intrinsic heart rate and corresponding core body temperature during the two days of autonomic inhibition. Using the $Q_{10}$ value 2, we corrected the intrinsic rate to the measured core body temperature (HR-Tc) for both sexes (see methods). At 37 °C, the HR-Tc was 445 bpm.

**ΔHR and autonomic regulation of heart rate.** To calculate ΔHR over the 24-hour cycle, we subtracted the HR-Tb from the experimentally measured heart rate. Figure 4A–C and Fig. 4G–I show the hourly averages of the experimentally measured heart rate, HR-Tb, and ΔHR for the two days before inhibition, during inhibition, and recovery in both female and male mice. Both sexes showed day vs. night differences in heart rate and ΔHR before inhibition and during recovery (Fig. 4D,F,J,I). Before inhibition and during recovery, ΔHR was greater than 0 at night, indicating higher sympathetic regulation in females. During the day, ΔHR was close to 0 in female mice, suggesting balanced autonomic regulation. In male mice, ΔHR was less than 0 during the day and close to 0 at night, indicating higher parasympathetic regulation during the day. Autonomic receptor inhibition caused ΔHR to approach 0 during day and night for both sexes (Fig. 4E,K). These findings demonstrate that ΔHR can be used to quantify day-night changes in the autonomic regulation of heart rate. Both sexes showed day vs. night differences in core body temperature before inhibition, during autonomic receptor inhibition, and during recovery (Supplementary Fig. 3).

**dLAN causes sex-specific differences in the day-night autonomic regulation of heart rate**
We calculated the HR-Tb and the ΔHR for the study described in Fig. 2C across the three different phases: mice housed in a 12-hour light/12-hour dark cycle with unrestricted food access; exposed to dLAN with unrestricted food access; and continued in dLAN but with feeding restricted to the dLAN cycle. Figure 5A–F show a seven-day time series of hourly mean heart rate, HR-Tc, and ΔHR from both female and male mice.

**HR-Tc.** We quantified the day-night difference in HR-Tc for each phase in female and male mice (Fig. 5G,J). There were minimal sex differences in how the HR-Tb changed across the three phases. The HR-Tc was higher at night than during the day in all phases. No differences were observed in the night HR-Tc across the phases. The day HR-Tb measured from mice undergoing dLAN with time-restricted feeding tended to be lowest. Cosinor analysis indicated that the amplitude of the 24-hour HR-Tc and ΔHR rhythm was smallest in mice exposed to dLAN with unrestricted food access (Fig. 5I,L).

**ΔHR.** We quantified the day-night difference in ΔHR for each phase in female and male mice (Fig. 5H,K). ΔHR was higher at night than during the day in all three phases. Cosinor analysis showed that the amplitude of the 24-hour ΔHR rhythm was lowest in mice exposed to dLAN with unrestricted food access (Fig. 5I,J). The day ΔHR measured across phases was lowest (most negative) for restricted food access under dLAN. There were notable sex-specific differences. For mice with unrestricted food access under dLAN, the night ΔHR was lowest (closest to 0) in female

mice, and the day ΔHR was highest (closest to 0) in male mice. These data showed that dLAN exposure with unrestricted food access caused sex-specific changes in autonomic regulation of the day-night heart rate rhythms. dLAN decreased the relative sympathetic regulation of heart rate in female mice at night (making it slower) and increased the relative sympathetic regulation of heart rate in male mice during the day (making it faster) (Fig. 5H,K). Directly comparing how the ΔHR changed during the day and night following exposure to dLAN with unlimited food access demonstrated a negative change in ΔHR for female mice at night and a positive change in ΔHR for male mice during the day (Supplementary Fig. 4). Time-restricted feeding to the dLAN cycle increased the relative sympathetic regulation in female mice at night and decreased the relative sympathetic regulation in both female and male mice during the day (Fig. 5H,K).

**dLAN causes sex-specific differences in the day-night blood pressure rhythms**

**Blood pressure.** dLAN with unrestricted access to food caused sex-specific differences in autonomic regulation of the heart rate. We tested whether these differences were associated with sex-specific differences in blood pressure across the three phases (i.e., (Fig. 2B). Figure 6A though 6F shows the corresponding seven-day time series of hourly mean systolic and diastolic blood pressures measured from female and male mice for each phase. Cosinor analyses revealed that the amplitude of the 24-hour blood pressure rhythms in female mice was not affected by exposure to dLAN (Fig. 6G,J). In contrast, the amplitudes of the 24-hour systolic and diastolic blood pressure rhythms were reduced in male mice exposed to dLAN with unrestricted food access compared to those in the 12-hour light/12-hour dark cycle. Restricting feeding to the dLAN cycle increased the amplitudes of the 24-hour systolic and diastolic rhythms in male mice to levels observed under the 12-hour light/12-hour dark cycle.

Exposure to dLAN with unrestricted food access also unmasked sex-specific differences in the mesor (Fig. 6H,K) and acrophase (Fig. 6I,L) of the blood pressure rhythms. Specifically, the mesor in systolic and diastolic blood pressure rhythms were higher in male mice, and the acrophase of the systolic blood pressure rhythm peaked later.

We tested how these sex-specific differences in the 24-hour regulation of blood pressure across the different phases translated into differences in systolic, diastolic, and mean arterial blood pressure measured during the day (Fig. 7A–C) or night (Fig. 7D–F). In female mice, there were no differences in the systolic, diastolic, or mean arterial pressure across the three phases during the day or night.

In male mice, exposure to dLAN with unrestricted food access increased the day systolic, diastolic, and mean arterial blood pressure compared to mice housed in 12-hour light/12-hour dark cycles. These differences in blood pressure were normalized after dLAN-restricted feeding. The night systolic, diastolic, and mean arterial pressure in male mice did not change across the three phases.

There were several notable sex-specific differences in day vs. night blood pressures. Compared to female mice, the day systolic, diastolic, and mean blood pressure were higher in the male mice exposed to dLAN with unrestricted food access (Fig. 7A–C). Also, the night systolic, diastolic, and

**A. Dim light at night effects on the heart rate and core body temperature**

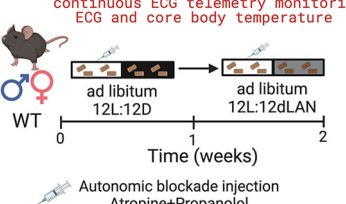

**B. Effects of time-restricted feeding during dim light at night**

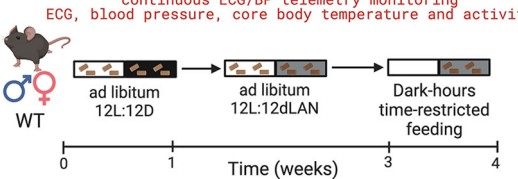

**C. The role of autonomic nervous system on the heart rate**

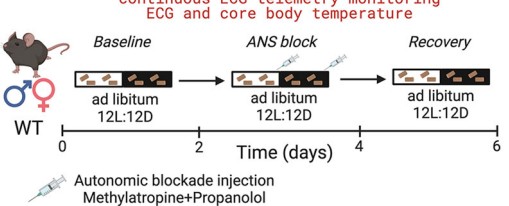

**Fig. 2 | Experimental Protocol.** All experiments were done under thermoneutrality (30 °C). **A** Dim night light effects on the heart rate and core body temperature. Male and female wild-type (WT) mice (n = 5–6/sex) were implanted with telemetry probes to continuously record heart rate (HR) and core body temperature (Tb). Mice were kept under 12-hour light and 12-hour dark cycles (LD; light: dark; 200 lux: 0 lux) with ad libitum access to food for one week. After initial acclimation, mice were exposed to 12-hour light and 12-hour dim light at night cycles (dLAN; light: dim light at night; 200 lux: 5 lux) for a week. To measure intrinsic heart rate, mice were injected with propranolol (10 mg/kg) and atropine sulfate (10 mg/kg) to inhibit β-adrenergic and muscarinic receptor signaling at zeitgeber time 6-7 at both LD and dLAN conditions. **B** Effects of time-restricted feeding during dim light at night. In another set of experiments, WT mice (n = 5–6/sex) were implanted with bipotential telemetry probes and exposed to LD for one week, followed by dLAN for two weeks and one week of food restriction to the dim light cycle. HR, blood pressure (BP), Tb, and activity were continuously monitored throughout the experiment. The dark and gray shades represent dark or dim light cycles, respectively. **C** The role of the autonomic nervous system in regulating heart rate. Male and female WT mice (n = 4–5/sex) were implanted with telemetry probes to record HR and Tb. Mice were injected intraperitoneally with propranolol (10 mg/kg) and methylatropine (1 mg/kg) at zeitgeber times 23.5 and 11.5 for two consecutive days to inhibit β-adrenergic and muscarinic receptor signaling. HR and Tb were continuously monitored throughout the experiment.

mean blood pressure were higher in the male mice compared to female mice after exposure to dLAN without or with restricted food access (Fig. 7D–F).

**Blood pressure dipping.** Blood pressure dipping refers to the decline in blood pressure that occurs during the night in people and day in mice, corresponding to the rest/sleep hours. In mice, it is measured as a percentage decrease in daytime blood pressure levels. Dipping >10% is considered healthy. We calculated the systolic, diastolic, and mean arterial blood pressure dipping percentage for all three phases (Fig. 7G–I). Mice housed in 12-hour light/12-hour dark cycles showed blood pressure dipping roughly 10%. Exposing mice to dLAN with unrestricted food access reduced dipping for all the blood pressure parameters. dLAN-cycle restricted feeding restored blood pressure dipping similar to levels observed in mice housed in 12-hour light/12-hour dark cycles. Although the changes in blood pressure

dipping across all three phases were similar for both sexes, the reasons for these changes differed. In female mice, the decreased blood pressure dipping following exposure to dLAN with unrestricted food access was driven by a reduction in the blood pressures measured at night, but in males, it was caused by an increase in the blood pressures measured during the day (Supplementary Fig. 5). This significant sex-specific difference in the loss of dipping mirrors autonomic changes in heart rate regulation: Female mice had a reduced sympathetic regulation of heart rate during the night, and males had increased sympathetic regulation of heart rate during the day.

### Time-restricted feeding to the dLAN cycle did not increase nighttime activity

**Activity.** We quantified the day-night difference in activity for each phase in both female and male mice (Fig. 2B, Supplementary Fig. 6). The activity levels were higher at night than during the day in all phases. No differences were observed in the day activity levels across the phases in both sexes. However, exposure to dLAN with or without time-restricted feeding tended to lower night activity levels compared to mice in 12-hour light/12-hour dark cycles. These data indicate that dLAN cycle-restricted feeding did not normalize activity levels. Cosinor analysis suggested the reduced nighttime activity levels in female mice were secondary to a decrease in the amplitude and phase shift in the 24-hour activity rhythms (Supplementary Fig. 6). Although not significant, a similar trend was seen in the phase of the 24-hour activity rhythms in male mice.

## Discussion

The present study examined the effects of dLAN on cardiovascular physiology in mice housed in thermoneutrality. dLAN decreased the amplitude of feeding, core body temperature, heart rate, blood pressure, and activity rhythms. We identified a central role for autonomic regulation of dLAN-induced sex-specific changes in heart rate. dLAN caused sex-specific changes in blood pressure, including higher daytime blood pressure in male mice and decreased blood pressure dipping in both male and female mice. Restricted feeding to the dim light cycle normalized many dLAN-induced changes to autonomic regulation of heart rate and blood pressure but not activity. The data demonstrate that dLAN-induced circadian disruption of cardiovascular physiology is improved by time-restricted feeding.

### dLAN impacts heart rate and core body temperature

Light input to the SCN drives day-night rhythms in feeding behavior, activity, core body temperature, and autonomic regulation of cardiovascular physiology (i.e., heart rate and blood pressure)[25–27]. However, more recent studies have shown that rhythmic feeding drives the phase and amplitude of day-night rhythms of core body temperature, heart rate, and blood pressure[17,18,26,28,29]. This raised the possibility that the dLAN-induced reduction in rhythmic feeding behavior caused the decreased amplitude in day-night core body temperature, heart rate, and blood pressure rhythms. Consistent with this possibility, we found that time-restricted feeding to the dLAN cycle increased the amplitude of the day-night core body temperature and heart rate rhythms to levels similar to mice housed in 12-hour light/12-hour dark cycles. However, time-restricted feeding to the dLAN cycle did not normalize the amplitude in activity rhythms. Thus, although time-restricted feeding to the dLAN cycle can reverse many changes to day-night core body temperature and cardiovascular rhythms, it does not appear to normalize the disruption in amplitude for all day-night rhythms.

### dLAN causes sex-specific differences in autonomic regulation of heart rate

Changes in the day-night heart rate and blood pressure rhythms caused by exposure to light at night are thought to be primarily mediated by autonomic signaling[14]. However, quantifying autonomic changes in the regulation of heart rate is complex. Heart rate variability is often used, but its interpretation remains controversial, and most heart rate variability studies

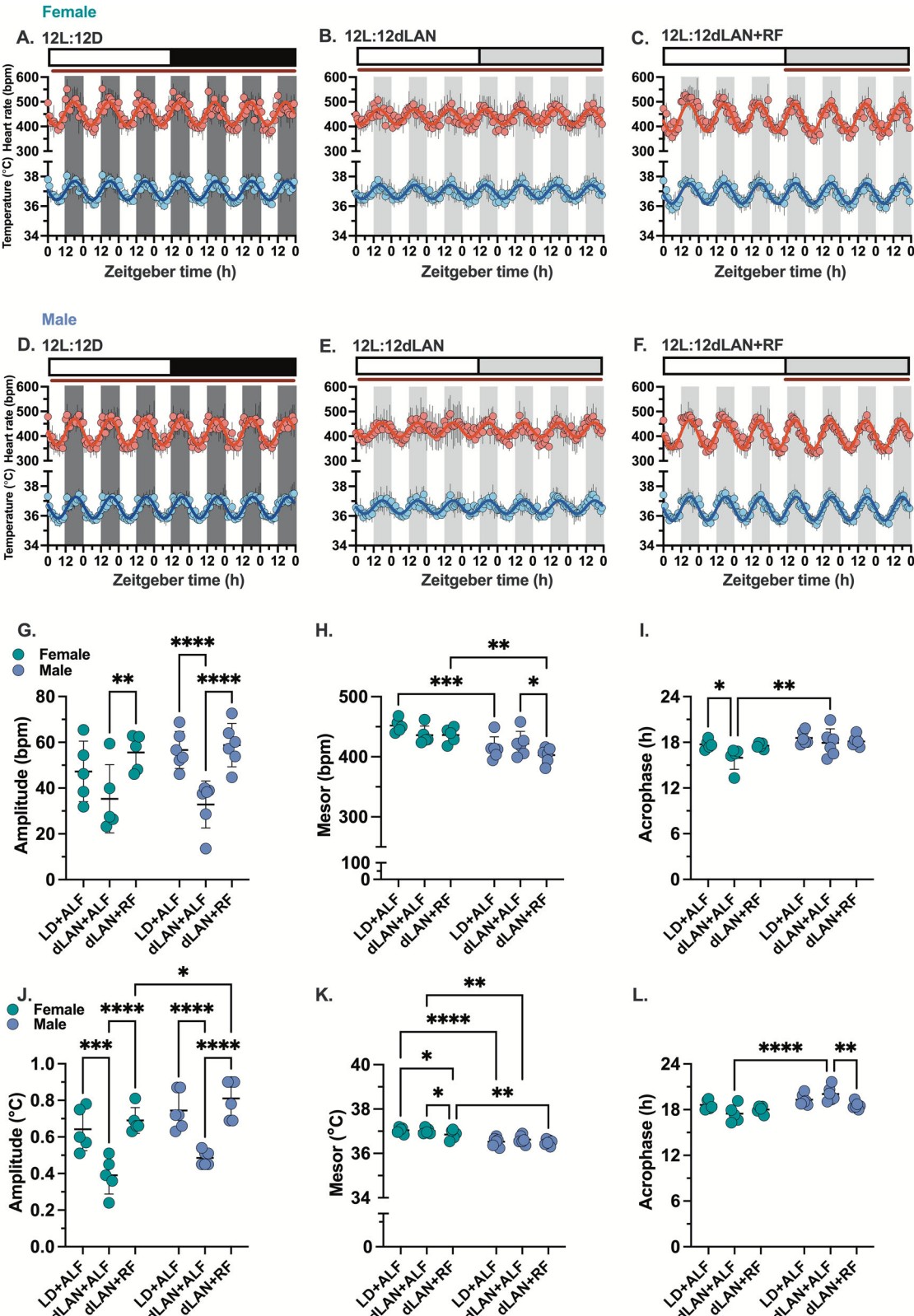

**Fig. 3 | Restricted feeding restores day-night heart rate and core body temperature rhythms.** The hourly mean for heart rate (HR, red) and core body temperature (Tb, blue) data recorded from female and male mice were plotted as a function of zeitgeber time for seven consecutive days. **A,D** Show the data measured from female and male mice housed in LD + ALF (12-hour light: 12 h dark; 200 lux: 0 lux; ad libitum feeding), **B,E** show the data measured from female and male mice housed in dLAN+ALF (12-hour light: 12-hour dim light at night; 200 lux: 5 lux; ad libitum feeding) and **C,F** show the data measured from female and male mice housed in dLAN+RF (12-hour light: 12 h dim light at night; 200 lux: 5 lux; dim light restricted feeding) ($n$ = 5–6/sex). The shaded regions in the graphs correspond to the dark or dim light cycles. The inset above each graph shows the food accessibility (brown line) as a function of the LD or dLAN cycle. The individual female and male mouse time series data for the HR and Tb were fit to a cosine wave to calculate the amplitude (**G,J**), mesor (**H,K**), and acrophase (**I,L**) for each condition. Data are presented as a scatter plot with the mean and SD. Significance was determined using a 2-way repeated measures ANOVA and Tukey's posthoc test. *$p < 0.05$, **$p < 0.01$, ***$p < 0.001$, and ****$p < 0.0001$.

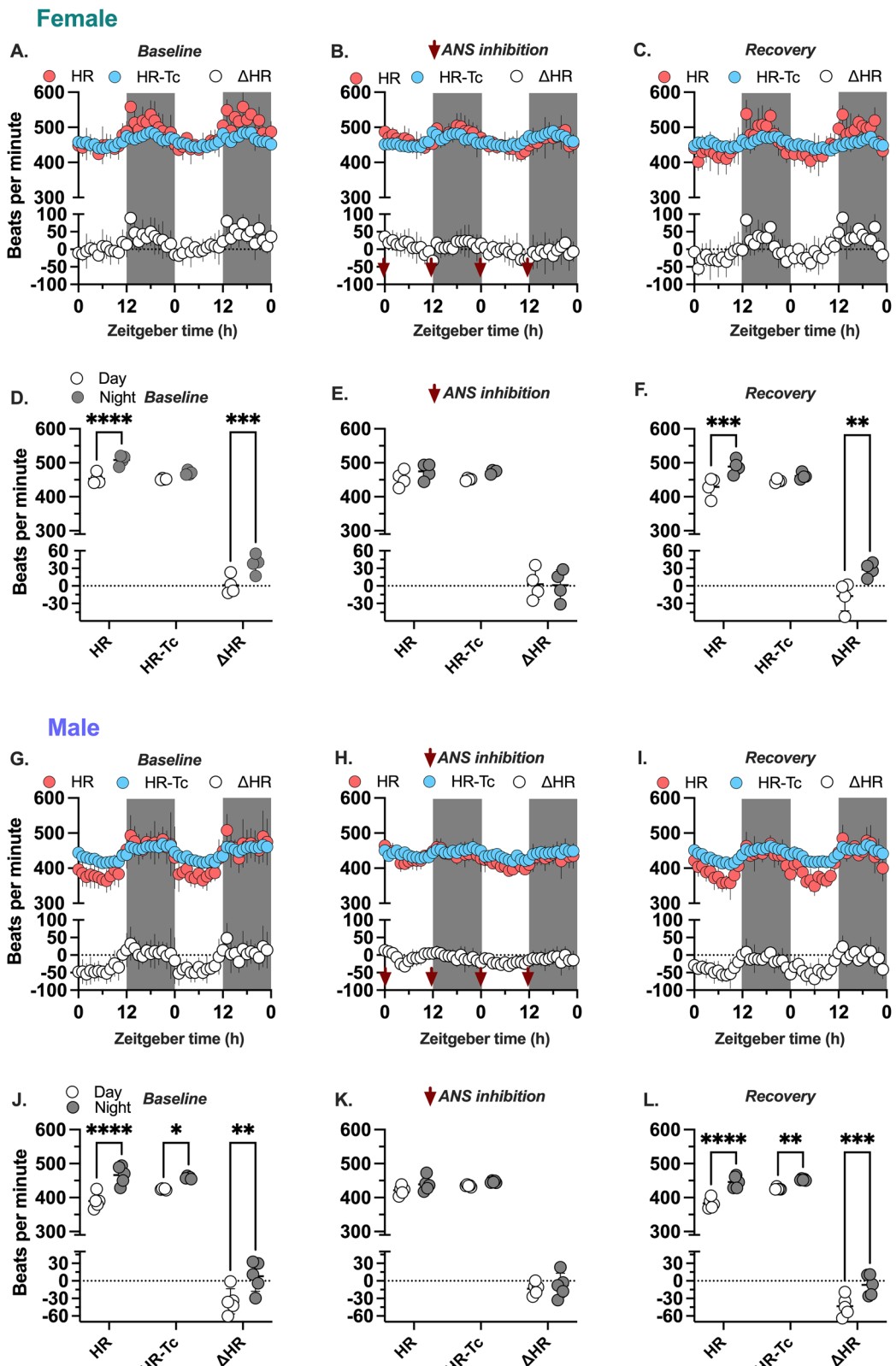

do not consider how changes in mean heart rate or core body temperature impact analyses[30,31]. Therefore, we combined autonomic receptor inhibition with core body temperature measurements to estimate how the intrinsic heart rate changes with core body temperature across the 24-hour cycle (HR-Tc). Calculating the difference between the experimentally measured heart rate and HR-Tc, i.e. ΔHR allowed us to quantify day-night rhythms in

autonomic regulation of heart rate. For the first time, we could quantify the daily rhythm in the ΔHR in female and male mice and how it is impacted by exposure to dLAN without or with time-restricted feeding. Despite dLAN decreasing the 24-hour rhythm in heart rate in female and male mice, the ΔHR analysis suggested it did so by distinct mechanisms. Specifically, exposure to dLAN reduced the relative sympathetic regulation of heart rate

**Fig. 4 | Quantification of day-night rhythms in autonomic regulation of heart rate at thermoneutrality.** The hourly mean for heart rate (HR, red), heart rate changed as a function of core body temperature (HR-Tc, blue), and heart rate associated with autonomic nervous system signaling (ΔHR, white) data recorded from female and male mice were plotted as a function of zeitgeber time for two consecutive days. **A**–**C** Show the data measured before inhibition (baseline), during autonomic nervous system (ANS) inhibition, and after recovery in female mice (n = 4). **D**–**F** The average HR, HR-Tc, and ΔHR data were measured for each mouse during the day (white circle) or night (gray circle) before inhibition (baseline), during ANS inhibition, and after recovery in female

mice. **G**–**I**. Show the data measured before inhibition (baseline), during ANS inhibition, and after recovery in male mice (n = 5). **J**–**L** The average HR, HR-Tc, and ΔHR data were measured for each mouse during the day (white circle) or night (gray circle) before inhibition (baseline), during ANS inhibition, and after recovery in male mice. The red arrow represents autonomic nervous system inhibition (ANS inhibition) injection at zeitgeber time 23.5 and 11.5 (lights on at ZT 0). Data are presented as a scatter plot with the mean and SD. Significance was determined using a 2-way repeated measures ANOVA and Tukey's posthoc test. *$p < 0.05$, **$p < 0.01$, ***$p < 0.001$, and ****$p < 0.0001$.

in female mice during the night but increased the relative sympathetic regulation of heart rate in male mice during the day.

## dLAN causes sex-specific differences in blood pressure

dLAN exposure in male mice with unrestricted food access increased the daytime systolic, diastolic, and mean arterial blood pressure. In contrast, in female mice, there was little difference in the 24-hour blood pressure rhythms or the absolute values of the systolic, diastolic, and mean blood pressure during the day or night. However, the impact that dLAN had on female blood pressure became clearer after calculating blood pressure dipping. Blood pressure dipping is an important clinical marker for cardiovascular disease in people. Generally, the blood pressure is lower during the night in people and during the day in mice[15,18,32,33]. Classically, a >10% decrease in resting blood pressure dipping is considered healthy, and the loss or decrease in blood pressure dipping has been suggested to be an index for future target organ damage and risk for cardiovascular events. Several studies previously explored the relationship between the rhythms of food intake and blood pressure in mice[18,26,29]. Similar to heart rate and core body temperature, feeding behavior modifies the amplitude and phase of blood pressure rhythms. Restricting food availability only to the light cycle abolishes normal blood pressure dipping in mice and induces "reversed" dipping in the db/db diabetic mouse models[18,26,29]. In contrast, restricting food availability to the dark cycle restores normal blood pressure dipping in db/db mice by reducing sympathetic signaling during the daytime while fasting[18]. Mice exposed to dLAN with unrestricted food access reduced blood pressure dipping in both female and male mice, and time-restricted feeding to the dLAN cycle restored blood pressure dipping to levels similar to that in 12-hour light and 12-hour dark cycles.

Reduction in female blood pressure dipping following exposure to dLAN with unrestricted food was related to a lower nighttime blood pressure trend. In contrast, in male mice, it was driven by the higher daytime blood pressure. This sex-specific difference in reduced blood pressure dipping mirrored the sex-specific differences in autonomic regulation of heart rate. Female mice showed a reduction in the relative sympathetic regulation of heart rate at night, and male mice showed an increase in the relative sympathetic regulation of heart rate during the day. In both cases, dLAN cycle-restricted feeding effectively restored autonomic regulation of heart rate and blood pressure dipping.

## Time-restricted feeding to the dLAN cycle did not restore the amplitude of activity rhythms

Light at night exposure can change locomotor activity in diurnal and nocturnal animals[7,16,34]. We found that exposing mice to dLAN reduced the amplitude of 24-hour activity rhythms in female mice and caused a reduction in activity during the dark cycle in both female and male mice. The decrease in activity during the dLAN cycle persisted even during dLAN cycle-restricted feeding. Thus, time-restricted feeding to the dLAN cycle did not increase or normalize the amplitudes for all day-night rhythms disrupted by exposure to dLAN.

## Implications

An important question is whether these results may be clinically relevant to humans. Day-night rhythms in the physiology and behavior of people differ from mice because people are diurnal, and mice are nocturnal. However, the

24-hour rhythms in heart rate, blood pressure, and core body temperature do not align with light-dark cycles but with the person or animal's activity and feeding cycles. So, like mice, day-night rhythms in heart rate, blood pressure, and core body temperature in people align with activity and feeding rhythms. Data suggest that light at night decreases the amplitude of day-night heart rate and blood pressure rhythms in people and small animals but through distinct mechanisms[14]. The mechanisms for these reductions are attributed to changes in autonomic signaling. In people, the reduction in daily heart rate and blood pressure rhythms is thought to be secondary to an increase in relative sympathetic signaling during their sleep cycle at night[14,35]. These results are qualitatively similar to what we observed in male mice exposed to dLAN. We found that exposure to dLAN increased the relative sympathetic regulation of heart rate and blood pressure during their daytime rest cycle.

The impact of artificial light at night on people and dLAN on male mice differs from that of artificial light at night on male rats housed at room temperature[14]. Light at night in rats appears to decrease the relative sympathetic signaling at night to decrease the amplitude of the 24-hour heart rate and blood pressure rhythms[13,36]. One possible reason for the difference between our studies in male mice and previous studies in rats is that we studied the effects of dLAN in mice housed at thermoneutrality to limit cold-induced sympathetic nervous system activation. Housing mice in thermoneutrality lowers their metabolic rate and increases parasympathetic tone to slow resting heart rate[19]. These conflicting results raise the intriguing possibility that the impact light at night has on autonomic signaling during the 24-hour cycle may depend on basal metabolic rate and autonomic tone.

We did not see an increase in the relative sympathetic regulation of heart rate and blood pressure during the day in female mice housed in thermoneutrality following dLAN exposure. Similar to male mice, housing female mice in thermoneutrality lowers their basal metabolic rate and heart rate. However, female mice tended to have a higher core body temperature than male mice, and studies show that female mice prefer warmer ambient temperatures[37,38]. The reasons for these differences are unclear, but they persist after gonadectomy, suggesting that gonadal hormones do not drive the higher metabolism in female mice[37]. This raises the possibility that resting metabolic rates may be a key determinant of whether artificial light at night and dLAN increase or decrease the relative sympathetic signaling during inactive or active cycles, respectively. Whether or not this observation translates to humans with different basal metabolic rates requires further study.

## Limitations

The concept that light at night decreases the amplitude in day-night rhythms of cardiovascular physiology is not new, nor is the concept that it is linked to increased cardiometabolic risk[4,10,13–15]. A limitation of this study is that we did not include an experimental group for nighttime restricted feeding in mice housed in 12-hour light and dark cycles. Previous studies suggest that nighttime restricted feeding in male mice housed in 12-hour light and dark cycles did not show significant changes in 24-hour blood pressure or heart rate rhythms[18]. Previous studies suggest that time-restricted feeding impacts the 24-hour heart rate, blood pressure, and core body temperature rhythms by altering autonomic signaling[17,18,29]. Our new data indicate that dLAN impacts the autonomic regulation of the heart in both female and male mice to disrupt 24-hour heart rate and blood pressure rhythms. Since time-restricted feeding impacts autonomic regulation of the

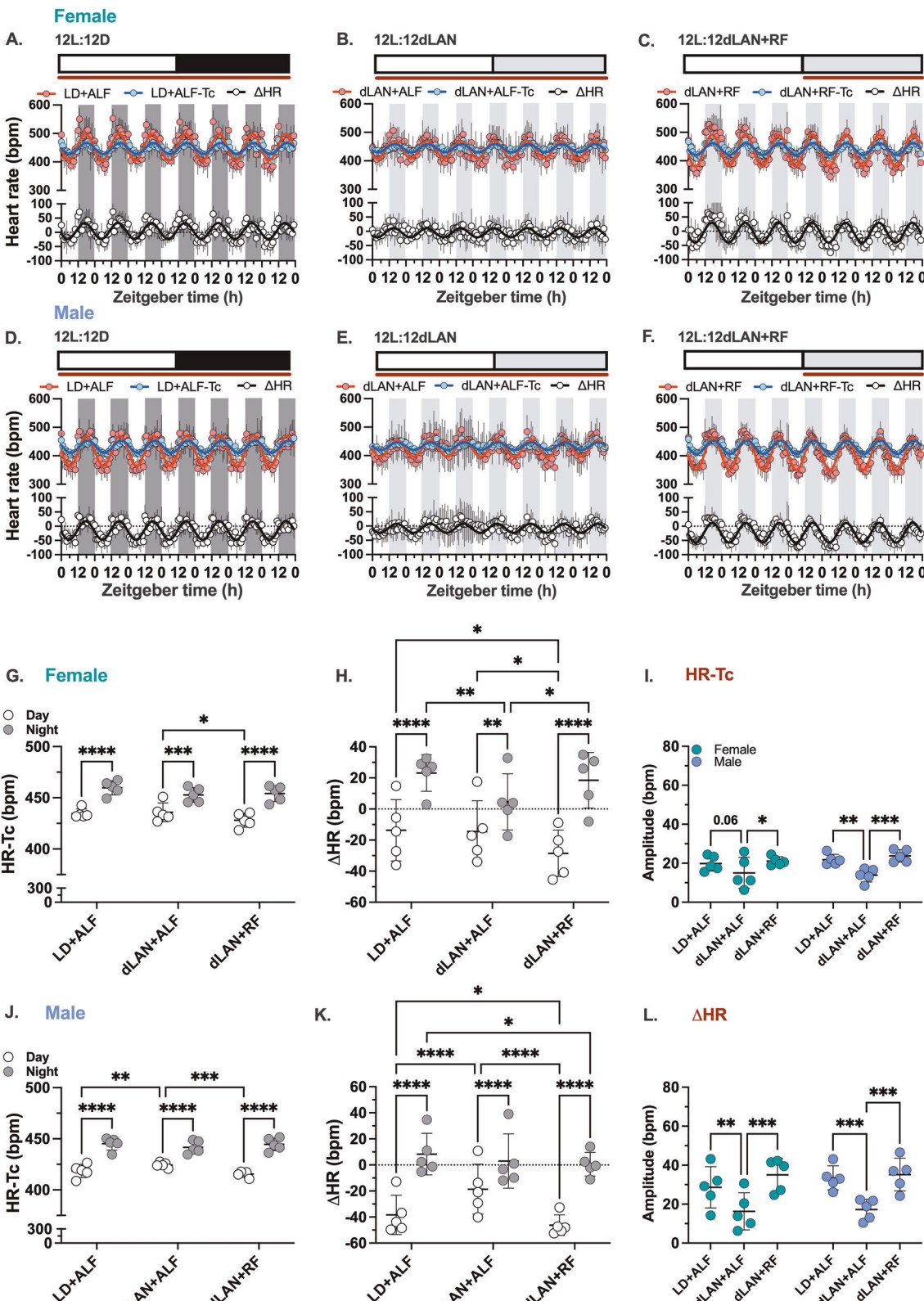

heart and blood pressure, we tested whether dLAN cycle-restricted feeding could normalize autonomic signaling to the heart and improve 24-hour heart rate and blood pressure rhythms. However, further study is required to determine whether or not the attenuation of food intake during dLAN or some other mechanism is responsible for disrupted autonomic signaling during dLAN. What is new in this study is that we performed the work at

thermoneutrality to limit confounds associated with thermogenesis, and we quantified the day-night rhythms in the autonomic regulation of heart rate (considering changes in daily core body temperature). This allowed us to identify sex-specific differences in the autonomic regulation of heart rate after exposure to dLAN. The idea that models of shift work impact circadian physiology in females and males differently is not new. Most recently, sexual

**Fig. 5 | dLAN causes sex-specific differences in the day-night autonomic regulation of heart rate.** The hourly mean for heart rate (HR, red), heart rate changed as a function of core body temperature (HR-Tc, blue), and heart rate associated with autonomic nervous system signaling (ΔHR, white) data recorded from female and male mice were plotted as a function of zeitgeber time for seven consecutive days. **A,D** Show the data measured from female and male mice housed in LD + ALF (12-hour light: 12 h dark; 200 lux: 0 lux; ad libitum feeding), **B,E** show the data measured from female and male mice housed in dLAN+ALF (12-hour light: 12-hour dim light at night; 200 lux: 5 lux; ad libitum feeding) and **C,F** show the data measured from female and male mice housed in dLAN+RF (12-hour light: 12 h dim light at night; 200 lux: 5 lux; dim light restricted feeding) (*n* = 5/sex). The shaded regions in the graphs correspond to the dark or dim light cycles. The inset above each graph shows the food accessibility (brown line) as a function of the LD or dLAN cycle. **G,J** The average HR-Tc data measured for each mouse LD + ALF, dLAN+ALF, and dLAN +RF during the day (white circles) or night (gray circles) in females and males. **H,K** In females and males, the average HR data was measured for each mouse LD + ALF, dLAN+ALF, and dLAN+RF during the day (white circles) or night (gray circles). The individual female and male mouse time series data for the HR, HR-Tc, and ΔHR were fitted to a cosine wave to calculate each condition's amplitude (I., L.). Data are presented as a scatter plot with the mean and SD. Significance was determined using a 2-way repeated measures ANOVA and Tukey's posthoc test. *$p < 0.05$, **$p < 0.01$, ***$p < 0.001$, and ****$p < 0.0001$.

dimorphism was seen in response to chronic circadian misalignment (a model of rotating shift work) on a high-fat diet[11]. This included identifying sex-specific differences in blood pressure dipping, with males being more affected than females. Our study focused on the acute effects of dLAN to understand the physiological consequences before the development of significant metabolic changes, and we provide mechanistic insight as it relates to sex-specific differences in sympathetic signaling during the day and night. We also found that intervening with time-restricted feeding could normalize dLAN's impact on cardiovascular health in both sexes. Although sex-specific differences were identified in this study, we did not perform gonadectomy studies. Future studies investigating the role of sex hormones and how they influence the autonomic response to dLAN are warranted but are currently beyond the scope of this work. This work lacks gene expression studies. Our data mechanistically implicate dLAN-induced reversible changes in the day-night autonomic regulation of the cardiovascular system as opposed to transcriptionally mediated mechanisms. It is not clear if the results of these studies extend to people. Future studies in people investigating the relationship between light exposure at night, feeding rhythms, time-restricted feeding, and autonomic heart rate and blood pressure regulation are needed.

## Conclusion
We showed that in mice housed at thermoneutrality, exposure to dLAN with unrestricted food access disrupts day-night rhythms in feeding, core body temperature, heart rate, and blood pressure parameters in both male and female mice. dLAN exposure unmasked sex-specific differences in the autonomic regulation of heart rate and blood pressure. Restricted feeding to the dim light cycle restored many disrupted day-night autonomic and cardiovascular rhythms but not activity. Time-restricted feeding may represent a chronotherapeutic strategy to mitigate the impact of light at night on cardiovascular physiology.

## Methods
All animal procedures complied with the Association for Assessment and Accreditation of Laboratory Animal Care guidelines with approval by the Institutional Animal Care and Use Committee (2019-3304) at the University of Kentucky. Wild-type SV129 female and male mice (*Mus musculus*) used in these studies were four to six months old. Mice were acclimated to thermoneutrality (30 ± 2 °C) in circadian boxes (Actimetrics) under 12-hour light (200 lux) and 12-hour dark (0 lux) cycles with ad libitum access to food and water for one week before starting experiments. For dLAN studies, the 12-hour dark cycle was replaced by a 12-hour dim light (5 lux) cycle (Fig. 2A,B). To initiate dim light cycle-restricted feeding, food was removed at the start of the light cycle (zeitgeber time 0 or ZT 0) and presented at the beginning of the dim light cycle (ZT 12).

### Feeding measurements
An automated feeder system (Phenome Technologies and Actimetrics) was used, and mice were fed pellets (average weight: 0.3 g/pellet) similar to regular chow (Dustless Precision Pellets, BioServ, F0175). ClockLab Chamber Control Software (Actimetrics) was used to control light cycles and record the amount and timing of food access[39].

### Heart rate, blood pressure, core body temperature, and activity measurements
Telemetry was used to measure electrocardiogram (ECG), blood pressure, core body temperature, and activity. Male and female mice were anesthetized with isoflurane and implanted with PhysioTel ETA-F10 (measure ECG, core body temperature, and activity) or PhysioTel HD-X11 (measures ECG, core body temperature, blood pressure, activity; Fig. 2B) telemetry transmitter units similar to that described previously[18,40]. Mice were allowed to recover for at least two weeks in 12-hour light/12-hour dark cycles with ad libitum access to food and water at room temperature (22 ± 1 °C). Data were recorded and analyzed using Ponemah software[18,41,42].

### Intrinsic heart rate measurements
We measured the heart rate after β-adrenergic and muscarinic receptor inhibition in male and female mice (i.e., the "intrinsic" heart rate). Mice were injected intraperitoneally with propranolol (10 mg/kg) and atropine sulfate (10 mg/kg) cocktail at ZT 6-7 in Experiment 1 (Fig. 2A)[17,24,43,44]. In another set of experiments, mice were injected intraperitoneally with propranolol (10 mg/kg) and methylatropine (1 mg/kg) at ZT 23.5 and ZT 11.5 for two consecutive days (Fig. 2C). We used methylatropine instead of atropine sulfate for the two-day autonomic inhibition experiment because atropine sulfate can affect the core body temperature in mice[45].

### Autonomic regulation of heart rate measurements
Studies show that heart rate changes with temperature with a $Q_{10}$ value of 2[23,46–48]. We quantified the average intrinsic heart rate and core body temperature in female and male mice during the two-day autonomic inhibition experiment (Fig. 2C). The average intrinsic rate and core body temperature allowed us to estimate how the intrinsic heart rate changes as a function of core body temperature with a $Q_{10}$ of 2 (HR-Tc):

$$HR\text{-}Tc = 445 \times (2)^{((Tb-37)/10)}$$

Taking the difference between the experimentally measured heart rate (HR) and the HR-Tc across the 24-hour cycle enabled us to quantify the change in heart rate associated with autonomic nervous system signaling (ΔHR).

$$\Delta HR = (HR) - (HR\text{-}Tc)$$

The ΔHR measured from female and male mice during the two-day autonomic inhibition experiment approached 0 (i.e., autonomic inhibition eliminated the ΔHR; See below Fig. 4B,E for females and 4H,K for males). ΔHR > 0 bpm indicated higher sympathetic regulation; ΔHR = 0 suggested balanced sympathetic and parasympathetic regulation; and ΔHR < 0 indicated higher parasympathetic regulation of heart rate. An increase in sympathetic regulation and a decrease in parasympathetic regulation are expected to increase ΔHR, and a decrease in sympathetic regulation and an increase in parasympathetic regulation are expected to decrease ΔHR. We describe changes in ΔHR as sympathetic regulation relative to parasympathetic regulation, such that an increase in ΔHR reflects an increase in

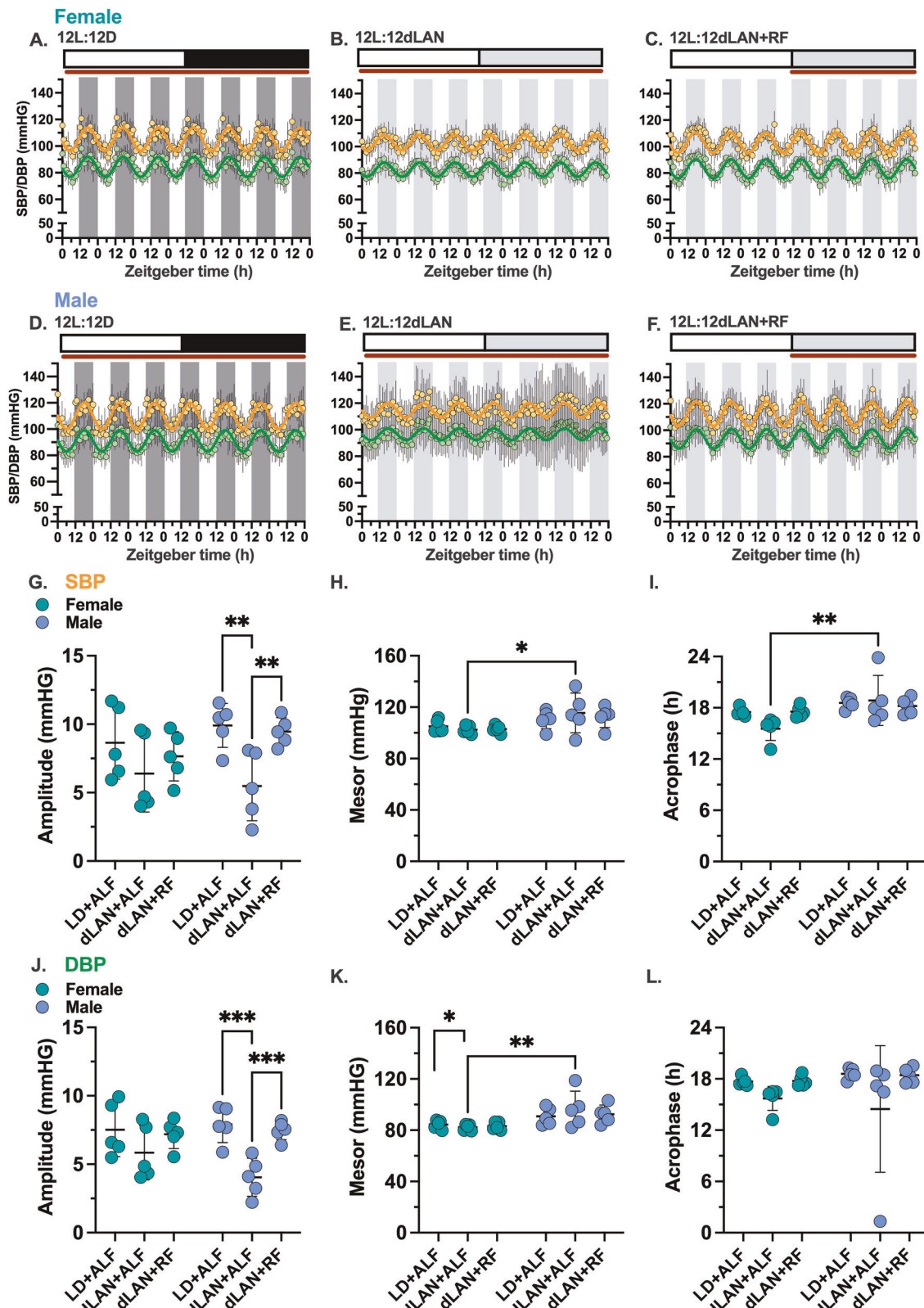

**Fig. 6 | dLAN reduced the amplitude of 24-hour blood pressure rhythm in male mice.** The hourly mean for systolic blood pressure (SBP, yellow) and diastolic blood pressure (DBP, green) data recorded from female and male mice were plotted as a function of zeitgeber time for seven consecutive days. **A,D** Show the data measured from female and male mice housed in LD + ALF (12-hour light: 12 h dark; 200 lux: 0 lux; ad libitum feeding), **B,E** show the data measured from female and male mice housed in dLAN+ALF (12-hour light: 12-hour dim light at night; 200 lux: 5 lux; ad libitum feeding) and **C,F** show the data measured from female and male mice housed in dLAN+RF (12-hour light: 12 hours dim light at night; 200 lux: 5 lux; dim light restricted feeding) (n = 5/sex). The shaded regions in the graphs correspond to the dark or dim light cycles. The inset above each graph shows the food accessibility (brown line) as a function of the LD or dLAN cycle. The individual female and male mouse time series data for the SBP and DBP were fit to a cosine wave to calculate the amplitude (**G,J**), mesor (**H,K**), and acrophase (**I,L**) for each condition. Data are presented as a scatter plot with the mean and SD. Significance was determined using a 2-way repeated measures ANOVA and Tukey's posthoc test. *$p < 0.05$, **$p < 0.01$ and ***$p < 0.001$.

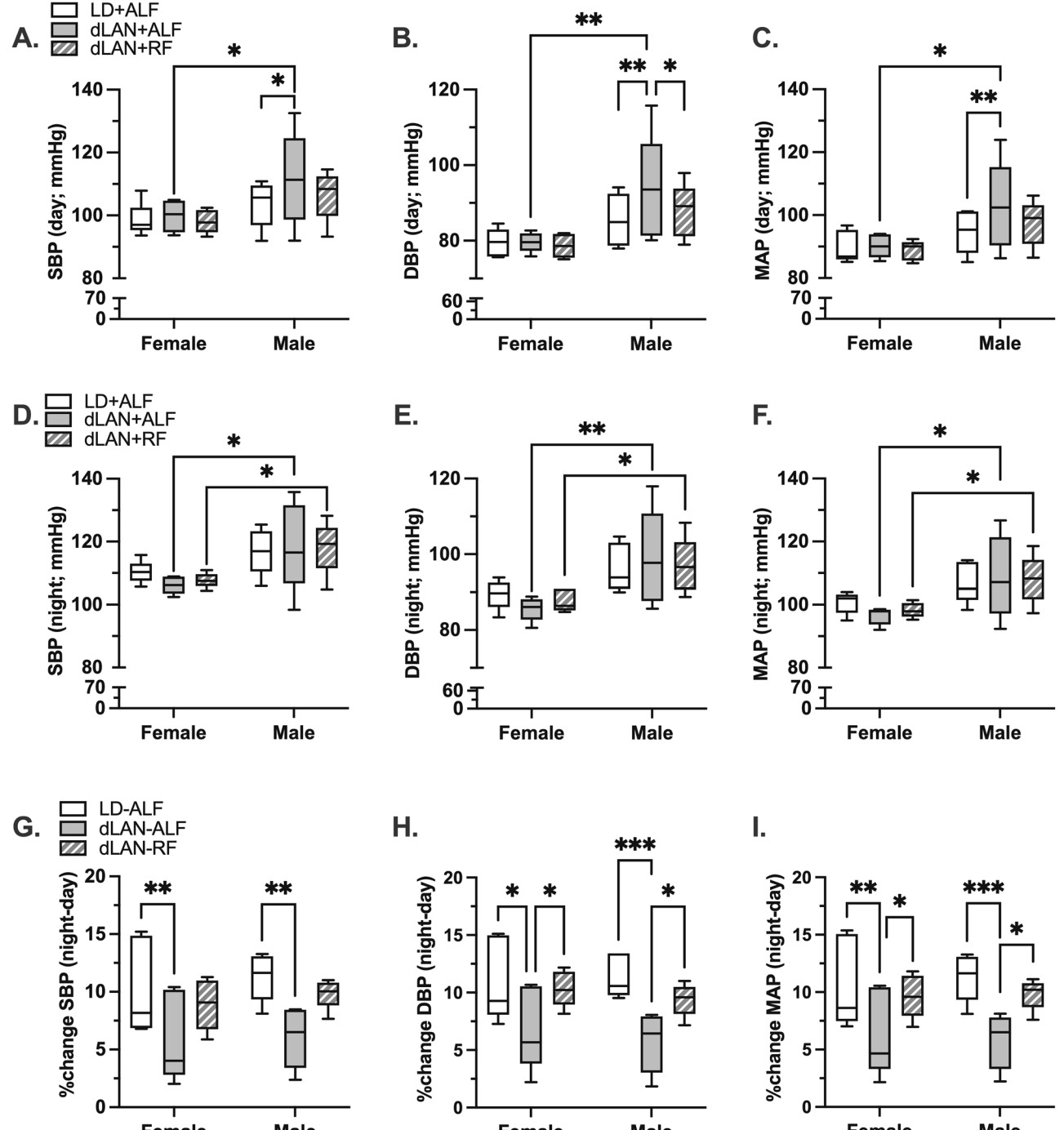

**Fig. 7 | dLAN increased daytime systolic, diastolic, and mean arterial blood pressure in male mice. A–C.** Show the average daytime systolic blood pressure (SBP) and diastolic blood pressure (DBP) and mean arterial pressure (MAP) variation, respectively, in female and male mice ($n = 5$/sex) under LD + ALF, dLAN +ALF, and dLAN+RF. **D–F** Show average nighttime SBP, DBP, and MAP variation in females and males under LD + ALF, dLAN+ALF, and dLAN+RF. **G–I** Show percent change as a measure of dipping in SBP, DBP, and MAP during the day hours compared to night hours in females and males. Data are presented as box-whisker plots (median) with error bars representing minimum and maximum values. Significance was determined using a 2-way repeated measures ANOVA and Tukey's posthoc test. $*p < 0.05$, $**p < 0.01$ and $***p < 0.001$.

the relative sympathetic regulation of heart rate and a decrease in ΔHR represents a decrease in the relative sympathetic regulation of heart rate.

**Statistics and reproducibility**
The hourly mean heart rate, blood pressure, and activity data were analyzed using Ponemah. The analysis was done blinded to the different experimental conditions. The Data are shown as the mean and standard deviation of the

mean (SD) and analyzed in PRISM (GraphPad 10.1.1, Boston, Massachusetts, USA) unless specified. Normality was tested using the Shapiro-Wilk test. Telemetry data were imported and analyzed using cosinor in the ClockLab Analysis Software (Actimetrics) to quantify the amplitude (one-half of the peak to trough), rhythm-adjusted mean (mesor), and acrophase (the time of the peak amplitude) for each mouse. The 24-hour rhythm in ΔHR was measured using a non-linear unimodal cosine wave function in

PRISM:

$$(y = A + [B*\cos(2\pi(x - C)/24)])$$

A, B, and C denote mesor, amplitude, and acrophase, respectively. The significance of rhythms was confirmed with F-statistics, based on model $R^2$, the number of predictors in the model, and the total sample size (https://www.danielsoper.com/statcalc/calculator.aspx?id=15;)[49,50]. Paired Student's t-test compared the differences between the two groups. Two-way repeated measures analysis of variance (two-way RM ANOVA) tested the effects of sex and light condition or time of the day unless specified. Posthoc testing used Tukey's or Sidak's, and $p < 0.05$ was considered statistically significant (Supplementary Data). ECG that could not be reliably analyzed due to noise artifacts were excluded.

## Reporting summary

Further information on research design is available in the Nature Portfolio Reporting Summary linked to this article.

## Data availability

The data that support the findings of this study are available from the corresponding author upon reasonable request.

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

## Acknowledgements

This work was supported by National Heart, Lung, and Blood Institute grants R01HL153042, R01HL141343, and R01HL172813. This publication was supported by the National Center for Research Resources and the National Center for Advancing Translational Sciences, National Institutes of Health, through Grant UL1TR001998. The content is solely the authors' responsibility and does not necessarily represent the official views of the National Institutes of Health. We thank Dr. Don Burgess (University of Kentucky) for providing feedback and discussion. Biorender was used to generate the illustration in Fig. 2.

## Author contributions

All authors contributed to drafting the work and revising it critically for important intellectual content, approved the final version of the manuscript, and qualified for authorship. B.P.D. and A.P. conceptualized and designed the study. A.P., T.S., and W.S. performed the surgery. A.P. and T.S. performed all the experiments. A.P., D.S., A.E., and I.S. performed analysis. B.P.D., A.P., M.C.G., and E.A.S. wrote the initial draft and revised the manuscript.

## Competing interests

The authors declare no competing interests.
