## [transparent peer review · Communications Biology]

Reviewers' comments:

Reviewer #1 (Remarks to the Author):

The purpose of the study by Prabhat et al was to test the hypothesis that 'exposure to dim light at night (dLAN) disrupts feeding rhythms, leading to detrimental changes in autonomic signaling and day-night heart rate and blood pressure rhythms.' [Abstract]. The authors report that mice exposed to dLAN exhibit attenuated 24-hr rhythms in heart rate, core body temperature, and blood pressure. Time restricted feeding increased the amplitude of many of these parameters. The authors also suggest that sex differences are observed with regards to the contribution of sympathetic regulation. The authors conclude by stating that 'dLAN induces sex specific changes in autonomic regulation of heart rate and blood pressure, and time restricted feeding may represent a chronotherapeutic strategy to mitigate the cardiovascular impact of light at night' [Abstract].

Overall, this is an interesting study, that reports a number of curious observations. During the review process, the following concerns arose.

Major Concerns

1) Apparent lack of an important experimental group. As mentioned in the manuscript, prior studies have reported that time-restricted feeding impacts several of the parameters investigated in the current study. Based on this information, it is somewhat surprising that the current study lacks the condition 'LD+RF'. This experimental group is useful, as it allows the reader to understand whether dLAN and RF have additive effects on the parameters reported (which would potentially indicate that dLAN and RF exert effects independently). If this were the case, then attenuation of food intake rhythms by dLAN may not be the mechanism by which dLAN affects heart rate, blood pressure, and body temperature rhythms.

2) Sex Differences. If the Reviewer understands the data presented in Figures 4 and 5 in the manuscript, during which sex-specific effects in autonomic regulation are investigated, it appears that no statistical analyses are utilized to compare males versus females. More specifically, data for males are presented on graphs that are separate from females, and as such, statistical comparisons are not shown. Instead, the text describes differences in patterns as 'notable sex-specific differences'. Given that sex differences are mentioned in the Title, Abstract, and Conclusions, it is recommended that statistical analyses be performed to validate their existence.

Specific Concerns

- 1) Line 26. It is somewhat surprising that the stated hypothesis in the Abstract does not mention sex differences. The Reviewer simply wishes to clarify whether this was an accidental omission.
- 2) Line 100. Please provide the manufacturer for the automated feeders.
- 3) Line 152. Were normality tests performed, so that it could be determined whether subsequent statistical analyses should be parametric or non-parametric?
- 4) Line 174. Please state the percentage of food consumed during the light versus dark (as opposed to saying 'most').
- 5) Figure 2. For the data presented in Figure 2, please show mesor and acrophase, in addition to amplitude.
- 6) Line 192. It is stated that 'Similar results were observed in male mice (Figure S1)'. When looking at Figure S1, no statistically significant differences are reported (yet statistically significant differences were reported for female mice in Figure 2, and are reported for male mice in Figure 3). As such, the Reviewer does not consider the data in Figure S1 to be similar.
- 7) Line 197. When making the statement 'intrinsic heart rate was not different in mice housed in 12-hour light/12-hour dark cycles or after exposure to dLAN (Figure S2)', what is the comparison to? Not different to what? Please clarify. Also, for the data in Figure S2, please show amplitude (plus mesor and acrophase).
- 8) Line 229. The statement 'The mesor of heart rate rhythms when housed under the 12-hour light/12-hour dark cycle and when the mice were exposed to dLAN with time restricted feeding to the dLAN cycle were higher in female mice (Figure 3H)' does not appear to be supported by the statistics shown in the figure. Other statements are made within this same paragraph that do not match statistical analysis presented in the Figures. Please review and clarify.

Reviewer #2 (Remarks to the Author):

This is a comprehensive and novel study assessing the physiological changes to cardiovascular function as a result of altered light-dark exposure. The authors conducted a thorough assessment of the changes to heart rate rhythm, blood pressure, body temperature, and food intake in mice exposed to a regular light-dark cycle, light-dim light cycle, and a third group where food was removed during the light cycle. Prabhat and colleagues demonstrated the alterations in the autonomic regulation of cardiovascular

function in female and male mice exposed to dim light at night (dLAN) using pharmacological approaches. Under thermoneutral conditions, female mice showed a reduction in sympathetic regulation during the dark period while an increase in sympathetic regulation was observed in male mice during the light phase following exposure to dLAN. There was a decrease in feeding during the day and core body temperature due to dLAN. The statistical analyses and comparisons were conducted properly.

Major Comments

The authors observed differential effects on heart rate rhythm and blood pressure due to autonomic regulation in female and male mice. What is absent is an explanation of the possible mechanisms of the sex differences. The Discussion section only reiterates the differences without providing the possible physiological or cellular basis for these observations. Have similar sex differences been reported in human subjects such as shift workers or people exposed to low-light ambient environments when sleeping? The additional statements would support the potential translational and clinical importance of the findings to the reported observations in human subjects and larger populations.

Minor Comments

Figure 1B

There is an unwanted label (“text”).

The asterisks are missing from the supplemental figures.

Reviewer #3 (Remarks to the Author):

The study was rigorously designed and performed and presents a significant amount of novel data on the impact of dim light at night (dLAN) on autonomic control of cardiovascular day-night rhythms. The authors report that although both sexes have decreased amplitude of the rhythmic changes in HR and BP with dLAN, the underlying autonomic mechanisms are different in males vs. females. As the authors state, this could have important implications for understanding sex-specific impacts of shift work in people.

I think I understand the basic approach here to separate the impact of core body temperature vs. autonomic regulation on HR (and overall, I think this is a strength of the study), but I don't quite understand why the authors focus on these changes being driven

by changes to sympathetic activity only (the authors use the phrase ‘sympathetic regulation’). A positive deltaHR could represent either an increase in sympathetic tone or a decrease in parasympathetic tone, or some combination thereof (and vice versa for negative deltaHR). Is there rationale as to why the authors suggest that these changes are solely due to changes in sympathetic regulation? Unless the studies are repeated using only sympathetic or parasympathetic blockade individually (which I am NOT suggesting they do – that is way beyond the scope of this study!), I don’t think the authors can definitively conclude that there were changes to either sympathetic or parasympathetic regulation, only the relative balance between the two branches or change in dominance. This does not necessarily impact the novelty or significance of the findings, but I think the authors should maybe be cautious with their interpretation and language regarding sympathetic and parasympathetic regulation vs. the balance between the two branches.

The other comment I have is that interpreting the impact of dLAN in mice, which are nocturnal, vs. humans is not necessarily intuitive and might be worth commenting on or discussing further. Presumably, dLAN in humans would disrupt the rest period, whereas dLAN in mice disrupts the active period – would we expect similar effects between mice and humans? Same for the time-restricted feeding – mice were fed only during dLAN, but presumably humans should be restricted to eating only in the light cycle. I think this would be worth a longer discussion and/or mentioning in the limitations of the study.

Minor comments:

Figure 1 is not really self-explanatory and could be improved. Why do some phases have little mice on top but others don’t? It would be helpful to mark the duration of each phase of the experiments rather than just the total duration. For example, for the time-restricted feeding study, how long was each phase of the study?

Supplemental Figure 1: there are no asterisks on panels B and C, should there be? Same for the rest of the supplementary figures.

Response to the Reviewers: We thank the reviewers for their constructive and careful review of our manuscript. We addressed each reviewer's questions, requests, and concerns in a point-by-point response and revised the manuscript accordingly. Incorporating the suggested modifications and addressing the reviewers' concerns has improved the overall clarity, significance, and impact of the manuscript.

Response to Reviewer 1:

Reviewer 1: Apparent lack of an important experimental group. As mentioned in the manuscript, prior studies have reported that time-restricted feeding impacts several of the parameters investigated in the current study. Based on this information, it is somewhat surprising that the current study lacks the condition 'LD+RF'. This experimental group is useful, as it allows the reader to understand whether dLAN and RF have additive effects on the parameters reported (which would potentially indicate that dLAN and RF exert effects independently). If this were the case, then attenuation of food intake rhythms by dLAN may not be the mechanism by which dLAN affects heart rate, blood pressure, and body temperature rhythms.

Response: The reviewer is correct that this experimental group may have helped determine if attenuation of food intake rhythms after exposure to dLAN is the mechanism by which dLAN affects heart rate, blood pressure, and body temperature rhythms. We now identify the lack of this experimental group as a limitation of our study.

"A limitation of this study is that we did not include an experimental group for nighttime restricted feeding in mice housed in 12-hour light and dark cycles. Previous studies suggest that nighttime restricted feeding in male mice housed in 12-hour light and dark cycles did not show significant changes in 24-hour blood pressure or heart rate rhythms¹⁸. Previous studies suggest that time-restricted feeding impacts the 24-hour heart rate, blood pressure, and core body temperature rhythms by altering autonomic signaling^{17,18,41}. Our new data indicate that the dLAN impacts the autonomic regulation of the heart in both female and male mice to disrupt 24-hour heart rate and blood pressure rhythms. Since time-restricted feeding impacts autonomic regulation of the heart and blood pressure, we tested whether dLAN cycle-restricted feeding could normalize autonomic signaling to the heart and improve 24-hour heart rate and blood pressure rhythms. However, further study is required to determine whether or not the attenuation of food intake during dLAN or some other mechanism is responsible for disrupted autonomic signaling during dLAN."

Reviewer 1: If the Reviewer understands the data presented in Figures 4 and 5 in the manuscript, during which sex-specific effects in autonomic regulation are investigated, it appears that no statistical analyses are utilized to compare males versus females. More specifically, data for males are presented on graphs that are separate from females, and as such, statistical comparisons are not shown. Instead, the text describes differences in patterns as 'notable sex-specific differences'. Given that sex differences are mentioned in the Title, Abstract, and Conclusions, it is recommended that statistical analyses be performed to validate their existence.

Response: We appreciate this suggestion, and we now include new data showing a significant difference in how day and night Δ HR changed following exposure to dLAN with unlimited access to food (see new Figure S4). To make a direct comparison between females and

males, we adjusted the calculation of the HR-Tc to use the same formula for both sexes: $HR-Tc = 445 \times (2)^{((Tb-37)/10)}$. We found that, when normalized for core body temperature, there was not a significant sex difference. We have updated the methods, Figures, and results using the modified calculation.

“We quantified the average intrinsic heart rate and core body temperature in female and male mice during the two-day autonomic inhibition experiment (**Figure 1C**). The average intrinsic rate and core body temperature allowed us to estimate how the intrinsic heart rate changes as a function of core body temperature with a Q_{10} of 2 (HR-Tc): $HR-Tc = 445 \times (2)^{((Tb-37)/10)}$ ”

We recalculated the ΔHR and updated the ΔHR in **Figures 4** and **Figure 5**.

To directly quantify the sex difference in response to the exposure to dLAN, we subtracted the ΔHR measured from mice after exposure to dLAN from the ΔHR measured from mice housed in a 12-hour light/12-hour dark cycle with unrestricted food access. New Figure S4 shows that exposure to dLAN with unrestricted food access increased the change ΔHR by about 20 beats per minute in male mice during the day and reduced the change ΔHR by about 20 beats during the night in female mice (see new **Figure S4** below). The changes in the ΔHR are consistent with an increase in the relative sympathetic regulation of heart rate in male mice during the day and a decrease in the relative sympathetic regulation of heart rate in female mice during the night following exposure to dLAN.

“ ΔHR : We quantified the day-night difference in ΔHR for each phase in female and male mice (**Figure 5H** and **5K**). ΔHR was higher at night than during the day in all three phases. Cosinor analysis showed that the amplitude of the 24-hour ΔHR rhythm was lowest in mice exposed to dLAN with unrestricted food access (**Figure 5I** and **5J**). The day ΔHR measured across phases was lowest (most negative) for restricted food access under dLAN. There were notable sex-specific differences. For mice with unrestricted food access under dLAN, the night ΔHR was lowest (closest to 0) in female mice, and the day ΔHR was highest (closest to 0) in male mice. These data showed that dLAN exposure with unrestricted food access caused sex-specific changes in autonomic regulation of the day-night heart rate rhythms. dLAN decreased the relative sympathetic regulation of heart rate in female mice at night (making it slower) and increased the relative sympathetic regulation of heart rate in male mice during the day (making it faster) (**Figure 5H** and **5K**). Directly comparing how the ΔHR changed during the day and night following exposure to dLAN with unlimited food access demonstrated a negative change in the ΔHR for female mice at night and a positive change in ΔHR for male mice during the day (**Figure S4**). Time-restricted feeding to the dLAN cycle increased the relative sympathetic regulation in female mice at night and decreased the relative sympathetic regulation in both female and male mice during the day (**Figure 5H** and **5K**).”

New Figure S4

Figure S4: dLAN causes sex differences in change in the Δ HR during the day and night.

Effects of dim light at night- The change in the Δ HR between dLAN+ALF and LD+ALF in females and males, respectively (n=5/sex). Data are presented as a scatter plot with the mean and SD. 2-way repeated measures ANOVA followed by Sidak's posthoc test for multiple comparisons. $p < 0.05$ is considered as statistically significant. ** $p < 0.01$ and *** $p < 0.001$.

Reviewer 1: Line 26. It is somewhat surprising that the stated hypothesis in the Abstract does not mention sex differences. The Reviewer simply wishes to clarify whether this was an accidental omission.

Response: Thank you for identifying this omission. We have modified the hypothesis statement in the abstract.

"We hypothesized that exposure to dim light at night (dLAN) disrupts feeding rhythms, leading to sex-specific changes in autonomic signaling and day-night heart rate and blood pressure rhythms."

Reviewer 1: Line 100. Please provide the manufacturer for the automated feeders.

Response: Thank you. We have provided the manufacturer's information for the automated feeders in the methods section.

"Feeding measurements: An automated feeder system (Phenome Technologies and Actimetrics) was used, and mice were fed pellets (average weight: 0.3 g/pellet) similar to regular chow (Dustless Precision Pellets, BioServ, F0175). ClockLab Chamber Control Software (Actimetrics) was used to control light cycles and record the amount and timing of food access²⁵."

Reviewer 1: Line 152. Were normality tests performed, so that it could be determined whether subsequent statistical analyses should be parametric or non-parametric?

Response: Thank you. We have tested normality through the Shapiro-Wilk test; hence, parametric statistical analysis was used. We have now mentioned this in the statistics section.

"Normality was tested using the Shapiro-Wilk test."

Reviewer 1: Line 174. Please state the percentage of food consumed during the light versus dark (as opposed to saying ‘most’).

Response: Thank you. We now mention the percentage % of food consumed.

“Mice under 12-hour light/12-hour dark cycles consumed most of their food at night, showing a significant day vs. night difference (percent food consumed: day- 21% vs. night- 79%). Exposure to dLAN increased food intake during the daytime (percent food consumed: day- 42% vs. night- 58%) but did not increase total 24-hour food intake (**Figure 2B**).”

Reviewer 1: Figure 2. For the data presented in Figure 2, please show mesor and acrophase, in addition to amplitude. Also, for the data in Figure S1, please show amplitude (plus mesor and acrophase).

Response: Thank you. We have now included mesor and acrophase in Figure 2 and similarly in Figure S1.

“Heart rate and core body temperature: The impact of dLAN on heart rate and core body temperature rhythms was determined using telemetry. Mice were kept under 12-hour light/12-hour dark cycles and then exposed to dLAN. **Figure 2C** shows the three-day time series of hourly mean heart rate and core body temperature data from female mice in 12-hour light/12-hour dark cycles and after exposure to dLAN for one week. Under 12-hour light/12-hour dark cycles, heart rate and core body temperature were lower during the day and higher at night (**Figure 2D and 2H**). Switching to dLAN eliminated the difference between day and night heart rate but not core body temperature. Cosinor analysis showed that dLAN decreased the day vs. night difference in heart rate by reducing the amplitude of the 24-hour heart rate rhythm (**Figure 2E**). Cosinor analysis also identified a trend towards a smaller amplitude in core body temperature rhythms (**Figure 2I**). dLAN did not

change the mesor (mean) of daily heart rate and core body temperature rhythms compared to mice in 12-hour light/12-hour dark cycles (**Figure 2F and 2J**). Compared to mice in 12-hour light/12-hour dark cycles, there was a trend for dLAN exposure to cause an earlier acrophase in the daily heart rate rhythm, and it resulted in an earlier acrophase in the daily core body temperature rhythm (**Figure 2F and 2J**). The impact of dLAN on heart rate and core body temperature rhythms was also determined in male mice using telemetry (**Figure S1**). We observed a qualitatively similar trend in dLAN's impact on the amplitude of the daily heart rate rhythm in male mice (**Figure S1C**). A notable difference between female and male mice was that exposure to dLAN did not advance the acrophase of the daily heart rate or core body temperature rhythm in male mice (compare **Figure 2F and 2J** with **Figure S1E and S1I**).”

“Figure 2: dLAN reduces the amplitude of feeding and heart rate rhythms in mice housed in thermoneutrality. A. The mean food intake measured every two hours from female mice housed in LD (12-hour light: 12-hours dark; 200 lux: 0 lux, half-filled circles) or dLAN (12-hour light: 12-hour dim light at night; 200 lux: 5 lux, grey circles) plotted as a function of zeitgeber time. The LD (red) and dLAN data (blue) were fitted with a cosine function **B.** The average food intake data was measured for each mouse housed in LD (half-filled circles) or dLAN (grey circles) during the day or night, and the amplitude of the day-night feeding rhythms was calculated using a cosine fit to the individual mouse data for each condition. **C.** The hourly mean heart rate (HR, red) or core body temperature (Tb, blue) data measured from female mice housed in LD or dLAN plotted as a function of zeitgeber time. The mean data were fitted with a cosine function (red and blue lines). **D.** The average HR measured for each mouse housed in LD (half-filled circles) or dLAN (grey circles) during the day or night. **E-G.** The amplitude, mesor, and acrophase of the day-night rhythms in HR were calculated using cosine fit to the individual mouse data for each condition. **H.** The average Tb was measured for each mouse housed in LD (half-filled circles) or dLAN (grey circles) during the day or night. **I-K.** The amplitude, mesor, and acrophase of the day-night rhythms in Tb were calculated using cosine fit to the individual mouse data for each condition. A paired t-test was used to compare the differences between the amplitudes. 2-way repeated measures ANOVA followed by Sidak’s posthoc test for LD and dLAN day vs. night comparisons. $p < 0.05$ is considered statistically significant. * $p < 0.05$, ** $p < 0.01$ and *** $p < 0.001$.”

Figure 1S: dLAN effects on the heart rate and core body temperature in male mice under thermoneutrality

A. The hourly mean heart rate (HR, red) or core body temperature (Tb, blue) data measured from male mice housed in LD or dLAN plotted as a function of zeitgeber time. The mean data were fitted with a cosine function (red and blue lines). **B.** The average HR measured for each mouse housed in LD (half-filled circles) or dLAN (grey circles) during the day or night. **C-E.** The amplitude, mesor, and acrophase of the day-night rhythms in HR were calculated using cosine fit to the individual mouse data for each condition. **F.** The average Tb was measured for each mouse housed in LD (half-filled circles) or dLAN (grey circles) during the day or night. **G-L.** The amplitude, mesor, and acrophase of the day-night rhythms in Tb were calculated using cosine fit to the individual mouse data for each condition. Data are presented as a

scatter plot with the mean and SD. 2-way repeated measures ANOVA followed by Sidak’s post hoc test for LD and dLAN day vs. night comparisons (**B, F**). Paired t-test compared the difference between the amplitudes (**C, G**), mesor (**D, H**), and acrophase (**E, I**). $p < 0.05$ is considered statistically significant. ** $p < 0.01$ and *** $p < 0.001$.

Reviewer 1: Line 192. It is stated that ‘Similar results were observed in male mice (Figure S1)’. When looking at Figure S1, no statistically significant differences are reported (yet statistically significant differences were reported for female mice in Figure 2 and are reported for male mice in Figure 3). As such, the Reviewer does not consider the data in Figure S1 to be similar.

Response: There were some similar trends in the change in the amplitude of the heart rates, but they did not reach significance. Since including the new data reporting the acrophase, we now observe new sex differences in the acrophase following exposure to dLAN. We have clarified our original statement and highlighted how exposure to dLAN differentially impacted the acrophase in female and male mice.

“The impact of dLAN on heart rate and core body temperature rhythms was also determined in male mice using telemetry (**Figure S1**). We observed a qualitatively similar trend in dLAN's impact on the amplitude of the daily heart rate rhythm in male mice (**Figure S1C**). A notable difference between female and male mice was that exposure to dLAN did not advance the acrophase of the daily heart rate or core body temperature rhythm in male mice (compare **Figure 2F** and **2J** with **Figure S1E** and **S1I**).”

Reviewer 1: Line 197. When making the statement ‘intrinsic heart rate was not different in mice housed in 12-hour light/12-hour dark cycles or after exposure to dLAN (Figure S2)’, what is the comparison to? Not different to what? Please clarify.

Response: Thank you. We have revised the statement for clarity.

“Intrinsic heart rate: We tested if dLAN altered heart rate by affecting intrinsic heart rate (i.e., heart rate after autonomic receptor inhibition) using telemetry. We measured the intrinsic heart rate in female and male mice under 12-hour light/12-hour dark cycles and after exposure to dLAN by injecting mice with atropine and propranolol at ZT 6-7. The intrinsic heart rate measured in the mice was generally higher than the pre-injection heart rate in mice housed in 12-hour light/12-hour dark cycles and after exposure to dLAN. Exposure to dLAN did not change the intrinsic heart rate compared to the intrinsic heart rate measured from the mice when they were housed in 12-hour light/12-hour dark cycles (**Figure S2**).

Reviewer 1: Line 229. The statement ‘The mesor of heart rate rhythms when housed under the 12-hour light/12-hour dark cycle and when the mice were exposed to dLAN with time restricted feeding to the dLAN cycle were higher in female mice (Figure 3H)’ does not appear to be supported by the statistics shown in the figure. Other statements are made within this same paragraph that do not match statistical analysis presented in the Figures. Please review and clarify.

Response: Thank you. We have modified the statement for clarity.

“Sex-specific differences existed in the mesor (**Figure 3H** and **3K**) and acrophase (**Figure 3I** and **3L**) of the 24-hour heart rate and core body temperature rhythms. When compared to male mice:

- (1) The mesor of heart rate rhythms when housed under the 12-hour light/12-hour dark cycle and when the mice were exposed to dLAN with time-restricted feeding to the dLAN cycle were higher in female mice (**Figure 3H**).
- (2) The mesor of core body temperature rhythms was higher in female mice when housed under the 12-hour light/12-hour dark cycle and when the mice were exposed to dLAN with unrestricted food access (**Figure 3K**).
- (3) When female mice were exposed to dLAN with unrestricted food access, the acrophase in heart rate (**Figure 3I**) and core body temperature rhythms (**Figure 3L**) occurred at an earlier time of day.”

Response to Reviewer 2:

Reviewer 2: The authors observed differential effects on heart rate rhythm and blood pressure due to autonomic regulation in female and male mice. What is absent is an explanation of the possible mechanisms of the sex differences. The Discussion section only reiterates the differences without providing the possible physiological or cellular basis for these observations. Have similar sex differences been reported in human subjects such as shift workers or people exposed to low-light ambient environments when sleeping? The additional statements would support the potential translational and clinical importance of the findings to the reported observations in human subjects and larger populations.

Response: Thank you for pointing this out. Although there are not many studies, we have related our findings to human and other animal studies in the discussion titled “Implications”

“Implications

An important question is whether these results may be clinically relevant to humans. Day-night rhythms in the physiology and behavior of people differ from mice because people are diurnal, and mice are nocturnal. However, the 24-hour rhythms in heart rate, blood pressure, and core body temperature do not align with light-dark cycles but with the person or animal’s activity and feeding cycles. So, like mice, the day-night rhythms in heart rate, blood pressure, and core body temperature in people align with their activity and feeding rhythms. Data suggest that light at night decreases the amplitude of day-night heart rate and blood pressure rhythms in people and small animals but through distinct mechanisms¹⁴. The mechanisms for these reductions are attributed to changes in autonomic signaling. In people, the reduction in the daily heart rate and blood pressure rhythms is thought to be secondary to an increase in relative sympathetic signaling during their sleep cycle at night^{14,47}. These results are qualitatively similar to what we saw in male mice. We found that exposure to dLAN increased the relative sympathetic regulation of heart rate and blood pressure during their rest cycle in the daytime.

The impact of artificial light at night on people and dLAN on male mice differs from that of artificial light at night on male rats housed at room temperature¹⁴. Light at night in rats appears to decrease the relative sympathetic signaling at night to decrease the amplitude of the 24-hour heart rate and blood pressure rhythms^{13,48}. One possible reason for the difference between our studies in male mice and previous studies in rats is that we studied the effects of dLAN in mice housed at thermoneutrality to limit cold-induced sympathetic nervous system activation. Housing mice in thermoneutrality lowers their metabolic rate and increases the parasympathetic tone to slow the resting heart rate¹⁹. These conflicting results raise the intriguing possibility that the impact light at night has on autonomic signaling during the 24-hour cycle may depend on basal metabolic rate and autonomic tone.

We did not see an increase in the relative sympathetic regulation of heart rate and blood pressure during the day in female mice housed in thermoneutrality following dLAN exposure. Similar to male mice, housing female mice in thermoneutrality lowers their basal metabolic rate and heart rate. However, the female mice tended to have a higher core body temperature than male mice, and studies show that female mice prefer warmer ambient temperatures^{49,50}. The reasons for these differences are unclear, but they persist after gonadectomy, suggesting that gonadal hormones do not drive the higher metabolism in female mice⁴⁹. This raises the possibility that resting metabolic rates may be a key determinant of whether artificial light at night and dLAN increase or decrease the relative sympathetic signaling during inactive or active cycles, respectively. Whether or not this observation translates to humans with different basal metabolic rates requires further study.”

Figure 1

A. Dim light at night effects on the heart rate and core body temperature

B. Effects of time-restricted feeding during dim light at night

C. The role of autonomic nervous system on the heart rate

Reviewer 2: Figure 1B There is an unwanted label (“text”).

Response: Thank you. We have fixed the typo.

Reviewer 2: The asterisks are missing from the supplemental figures.

Response: Thank you. We have corrected this.

Response to Reviewer 3:

Reviewer 3: I think I understand the basic approach here to separate the impact of core body temperature vs. autonomic regulation on HR (and overall, I think this is a strength of the study), but I don't quite understand why the authors focus on these changes being driven by changes to sympathetic activity only (the authors use the phrase ‘sympathetic regulation’). A positive deltaHR could represent either an increase in sympathetic tone or a decrease in parasympathetic tone, or some combination thereof (and vice versa for negative deltaHR). Is there rationale as to why the authors suggest that these changes are solely due to changes in sympathetic regulation? Unless the studies are repeated using only sympathetic or parasympathetic blockade individually (which I am NOT suggesting they do – that is way beyond the scope of this study!), I don't think the authors can definitively conclude that there were changes to either sympathetic or parasympathetic regulation, only the relative balance between the two branches or change in dominance. This does not necessarily impact the novelty or significance of the findings, but I think the authors should maybe be cautious with their interpretation and language regarding sympathetic and parasympathetic regulation vs. the balance between the two branches.

Response: We agree. We have modified the manuscript to include a discussion on this point in the methods and use the term relative sympathetic regulation as it relates to the change in the balance between sympathetic and parasympathetic regulation.

“The Δ HR measured from female and male mice during the two-day autonomic inhibition experiment approached 0 (i.e., autonomic inhibition eliminated the Δ HR; See below **Figure 4B, 4E** for females and **4H, 4K** for males). Δ HR > 0 bpm indicated higher sympathetic regulation; Δ HR = 0 suggested balanced sympathetic and parasympathetic regulation; and Δ HR < 0 indicated higher parasympathetic regulation of heart rate. An increase in sympathetic regulation and a decrease in parasympathetic regulation are expected to increase Δ HR, and a decrease in sympathetic regulation and an increase in parasympathetic regulation are expected to decrease Δ HR. We describe changes in Δ HR as sympathetic regulation relative to parasympathetic regulation, such that an increase in Δ HR reflects an increase in the relative sympathetic regulation of heart rate and a decrease in Δ HR represents a decrease in the relative sympathetic regulation of heart rate.”

Reviewer 3: The other comment I have is that interpreting the impact of dLAN in mice, which are nocturnal, vs. humans is not necessarily intuitive and might be worth commenting on or discussing further. Presumably, dLAN in humans would disrupt the rest period, whereas dLAN in mice disrupts the active period – would we expect similar effects between mice and humans? Same for the time-restricted feeding – mice were fed only during dLAN, but presumably humans should be restricted to eating only in the light cycle. I think this would be worth a longer discussion and/or mentioning in the limitations of the study.

Response: Thank you- this is an important point. We now discuss this in the Implications section and list it as a limitation of the study.

In the implications section.

“Implications

An important question is whether these results may be clinically relevant to humans. Day-night rhythms in the physiology and behavior of people differ from mice because people are diurnal, and mice are nocturnal. However, the 24-hour rhythms in heart rate, blood pressure, and core body temperature do not align with light-dark cycles but with the person or animal’s activity and feeding cycles. So, like mice, the day-night rhythms in heart rate, blood pressure, and core body temperature in people align with their activity and feeding rhythms. Data suggest that light at night decreases the amplitude of day-night heart rate and blood pressure rhythms in people and small animals but through distinct mechanisms¹⁴. The mechanisms for these reductions are attributed to changes in autonomic signaling. In people, the reduction in the daily heart rate and blood pressure rhythms is thought to be secondary to an increase in relative sympathetic signaling during their sleep cycle at night^{14,47}. These results are qualitatively similar to what we saw in male mice. We found that exposure to dLAN increased the relative sympathetic regulation of heart rate and blood pressure during their rest cycle in the daytime.”

In the limitations section.

“Our data mechanistically implicate dLAN-induced reversible changes in the day-night autonomic regulation of the cardiovascular system as opposed to transcriptionally mediated mechanisms. It is not clear if the results of these studies extend to people. Future studies in people investigating the relationship between light exposure at night, feeding rhythms, time-restricted feeding, and autonomic heart rate and blood pressure regulation are needed.”

Reviewer 3: Figure 1 is not really self-explanatory and could be improved. Why do some phases have little mice on top but others don't? It would be helpful to mark the duration of each phase of the experiments rather than just the total duration. For example, for the time-restricted feeding study, how long was each phase of the study?

Response: Thank you. We have revised Figure 1 accordingly and removed the mice from the figure. We have also revised the Figure 1 legend explaining the timelines of the experiments.

Figure 1

A. Dim light at night effects on the heart rate and core body temperature

B. Effects of time-restricted feeding during dim light at night

C. The role of autonomic nervous system on the heart rate

Reviewer 3: Supplemental Figure 1: there are no asterisks on panels B and C, should there be? Same for the rest of the supplementary figures.

Response: Thank you. We have modified Supplementary Figure 1 to include the missing p-value and statistics. Additional modifications were made in response to reviewer 1.

Supplementary Figure S1

Figure S1

REVIEWERS' COMMENTS:

Reviewer #1 (Remarks to the Author):

No additional comments - all previous comments were addressed.